# eDQA: Efficient Deep Quantization of DNN Activations on Edge Devices

**Wenhao Hu**                                                                          *2692597H@student.gla.ac.uk*
*School of Computing Science*
*University of Glasgow, Scotland, United Kingdom*

**Jude Haris**                                                                          *Jude.Haris@glasgow.ac.uk*
*School of Computing Science*
*University of Glasgow, Scotland, United Kingdom*

**Paul Henderson**                                                                    *Paul.Henderson@glasgow.ac.uk*
*School of Computing Science*
*University of Glasgow, Scotland, United Kingdom*

**José Cano**                                                                          *Jose.CanoReyes@glasgow.ac.uk*
*School of Computing Science*
*University of Glasgow, Scotland, United Kingdom*

**Reviewed on OpenReview:** *https://openreview.net/forum?id=SEIBCdgE5W*

## Abstract

Quantization of Deep Neural Network (DNN) activations is a commonly used technique to reduce compute and memory demands during DNN inference, which can be particularly beneficial on resource-constrained edge devices. To achieve high accuracy, existing methods for quantizing activations rely on complex mathematical computations or perform extensive online searches for the best hyperparameters. However, these expensive operations are impractical on edge devices with limited computational capabilities, memory capacities, and energy budgets. Furthermore, many existing methods either do not focus on sub-6-bit (or deep) quantization, or leverage mixed-precision approaches to achieve deep quantization on average but without further improving the hardware usage efficiency. To fill these gaps, in this paper we propose eDQA (Efficient Deep Quantization of DNN Activations on Edge Devices), a hardware-friendly method that focuses on sub-6-bit quantization of activations and leverages simple shifting-based operations and data compression techniques to achieve high efficiency and accuracy. We evaluate eDQA with 3, 4, and 5-bit quantization levels and four different DNN models on two different datasets. eDQA improves accuracy by 86.63 percentage points compared with four existing methods: direct quantization, classic power-of-two quantization, EasyQuant and the state-of-the-art NoisyQuant for sub-6-bit quantization. Additionally, we compare eDQA with NoisyQuant on an edge FPGA, achieving up to 309× speedup. The code is available at https://github.com/gicLAB/eDQA.

## 1 Introduction

Quantization is a popular compression technique to reduce the compute and memory demands of Deep Neural Networks (DNNs). During inference, DNN weights are typically fixed, so they can be quantized offline. However, activations are generated dynamically at runtime, which means that effective quantization methods must dynamically quantize activations online.

There are many methods for quantizing activations. The most straightforward one is to directly use the mathematical definition of quantization (i.e., simply apply the basic quantization formulas to the values to

be quantized) (Gholami et al., 2022), but it can provide limited accuracy. To obtain higher accuracy, more sophisticated approaches are used. For example, NoisyQuant (Liu et al., 2023) searches noise online and injects it in the activations before quantization and removes the noise after de-quantization (the opposite operation to quantization). However, these methods are not specifically designed for resource-constrained edge devices, where the computational cost of expensive mathematical operations (e.g. matrix multiplication) or large online search spaces is impractical. Furthermore, many of these methods (Liu et al., 2023; Xiao et al., 2023) do not focus on deep quantization for activations (e.g., quantizing to less than 6 bits), which is preferred for edge devices with limited memory; others use mixed-precision quantization (Zhao et al., 2024) to achieve deep quantization on average, which can cause inefficiencies for edge devices such as storage and bus bandwidth waste.

To fill these gaps, in this paper, we propose eDQA (Efficient Deep Quantization of DNN Activations on Edge Devices), a hardware-friendly method to deeply quantize DNN activations that yields higher accuracy than four existing methods (improves up to 86.63 percentage points). eDQA first determines the *importance* of each activation channel offline using training or calibration datasets. For important channels, eDQA first quantizes their values with $m$ extra bits, and then right-shifts the $m$ bits to achieve the target number of bits while storing the shifting errors in compressed form. During the de-quantization phase, the shifting errors are decompressed and added back to the corresponding channels. For unimportant channels, eDQA uses a direct method to quantize and de-quantize activations, i.e., it simply applies the mathematical definition of quantization/de-quantization (Gholami et al., 2022; Lin et al., 2024). By learning important channels offline, eDQA avoids searching during DNN inference. In addition, by using $m$ more bits, eDQA can quantize important channels with a different number of bits (i.e., mixed-precision quantization (Chen et al., 2024; Rakka et al., 2022)), reducing the quantization error exponentially, and efficiently shift back to the target bits without expensive mathematical computations. It is also important to note that: i) by right-shifting, all the quantized values can ultimately be stored using the same bit length, thus avoiding storage (Liu et al., 2024) and bus bandwidth waste; ii) by applying data compression to the shifting errors, eDQA can reduce the extra memory overhead from using the $m$ more bits. Separate storage of the deeply quantized activation values (with equal bit-lengths) and the shifting errors can achieve hardware benefits such as efficient storage and efficient bus bandwidth usage, while still providing high accuracy.

We evaluate the accuracy of eDQA with three sub-6-bit quantization levels (3, 4, and 5 bits) on four DNN models and two datasets (ResNet-32 (He et al., 2016) and MobileNetV2 (Howard et al., 2017) on CIFAR-10 (Krizhevsky, 2009); ResNet-18 (He et al., 2016) and ViT-b-16 (Dosovitskiy et al., 2021) on TinyImageNet (Le & Yang, 2015)). Additionally, we test eDQA on an edge device with an FPGA to demonstrate the feasibility of implementation on edge devices, the low resource utilization, and on-edge performance in terms of latency.

Overall, eDQA improves accuracy up to 86.63 percentage points compared with four existing methods: direct quantization, power-of-two (PoT) quantization (Vogel et al., 2019), EasyQuant (Wu et al., 2020), and the state-of-the-art method NoisyQuant (Liu et al., 2023). Furthermore, our implementation of eDQA on the edge FPGA outperforms NoisyQuant on the same hardware, achieving up to 309× speedup.

The main contributions of this paper are as follows:

- We propose eDQA, an efficient deep (sub-6-bit) quantization method for DNN activations on edge devices. eDQA deals with important activation channels separately and uses data compression techniques to optimize memory usage.

- We explore the patterns of shifting errors (generated by the shifting operation in eDQA) to justify their compressibility. We also compare four compression techniques (Huffman coding, Deflate, LZMA, and ZSTD) with respect to compression ratio and latency.

- We conduct a detailed evaluation of eDQA and compare it with three existing methods, direct quantization, PoT (Vogel et al., 2019) quantization and NoisyQuant (Liu et al., 2023), and find that eDQA substantially (by up to 75%) outperforms them on accuracy.

- We explore the impact of the hyperparameters of eDQA to provide more insights on the use of our method.

- We validate eDQA on an FPGA-based hardware accelerator to demonstrate the feasibility of implementation on edge devices, evaluate performance in terms of latency, and measure hardware resource utilization. We find that eDQA is up to 309× faster than NoisyQuant.

## 2 Background and Related Work

### 2.1 Quantization Methods

Quantization is a widely used compression method to reduce the precision format of parameters and activations in DNNs, which reduces memory storage and computational requirements (Gholami et al., 2022; Jacob et al., 2018). The initial DNN parameters, typically using a floating-point format (e.g., 32 bits), are converted into fixed-point or integer values that require fewer bits (e.g., 16 or 8 bits). The process normally includes clipping and rounding operations. Clipping restricts the minimum and maximum values of the quantization, whereas rounding approximates values to the nearest integers, which causes rounding errors that cannot be recovered during de-quantization. De-quantization is the opposite operation of quantization that approximately recovers the quantized values back to floating point.

Quantization methods are typically divided into uniform and non-uniform. In uniform quantization, the quantized values are evenly spaced, while in non-uniform quantization they are not (Gholami et al., 2022) (i.e., spaced according to a non-uniform function). Uniform quantization can be further divided into symmetric and asymmetric. In symmetric quantization, the clipping range is symmetric with respect to the origin, while in asymmetric quantization it is not (Gholami et al., 2022) (e.g., the clipping range can be skewed depending on the data). In this paper, we use uniform symmetric quantization due to its popularity and relatively low cost of computation (Gholami et al., 2022), and we define the direct uniform symmetric quantization and de-quantization as the **Direct** method (note that this is the same as the direct quantization method for unimportant channels mentioned in Section 1).

It is important to note that keeping the same bit length for the quantized values across the whole DNN model is straightforward, but can be sub-optimal for accuracy (Rakka et al., 2022). To improve this, many methods apply mixed-precision quantization that assigns different bit lengths to different parts of a DNN model, such as layers or channels, to satisfy their different precision sensitivities (Chen et al., 2024; Rakka et al., 2022). To achieve greater efficiency, Atom (Zhao et al., 2024) reorders the different groups of different quantized precisions to maintain the regular memory access. However, since the differences in bit length among the quantized values are not hardware friendly, mixed-precision quantization can cause inefficiencies such as storage (Liu et al., 2024) or bus bandwidth waste. Our eDQA method adapts mixed-precision quantization in an effective way (see Section 3), where quantized activations are stored using the same bit length while storing extra information separately.

Furthermore, previous works on DNN quantization apply quantization to the model weights and/or activations. For weights, AWQ (Lin et al., 2024) checks the importance of the weight channels for each layer and scales the important channels before quantization to reduce rounding errors. For activations, quantization is difficult because of its dynamic nature, i.e., the activation values are typically not known before inference. Therefore, to achieve high accuracy, expensive mathematical operations (e.g., matrix multiplication) or expensive online search spaces to find the best hyperparameters are required. For example, NoisyQuant (Liu et al., 2023) injects noise, which is obtained by online search within the given search spaces, to the activations before quantization and removes the noise after de-quantization. However, these previous methods do not focus on deep quantization like sub-6-bit quantization, thus being less suitable for resource-constrained edge devices. Atom (Zhao et al., 2024) tries to use mixed-precision quantization to archive deep quantization on average. However, as we mentioned previously, this method still faces hardware inefficiencies (such as storage and bus-bandwidth waste) caused by the bit-length differences of the values. Note that eDQA quantizes only activations, providing greater practical flexibility than methods that jointly quantize both weights and activations.

For edge devices, DNN quantization is very important due to the limited on-board resources (Tasci et al., 2025). FPGAs (Field Programmable Gate Arrays) as a flexible hardware platform have received more attention in recent years for quantized DNNs (Chang et al., 2021; Tasci et al., 2025). This is because FPGAs allow precise control over the bit-length used for on-chip memory, bus bandwidth, and compute elements, enabling hardware developers to efficiently design new accelerators that take advantage of low-bit DNN quantization methods.

### 2.2 Data Compression Techniques

As we introduced in Section 1, eDQA involves data compression techniques. Data compression aims to reduce the size of stored data using various methods. Huffman coding (Van Leeuwen, 1976) is a classic method of data compression. In Huffman coding, each character in the data is counted to get the frequency. Then, the characters that have higher frequency are encoded with shorter code while the characters that have lower frequency are encoded with longer code. Overall, when frequencies of different characters are distributed unevenly, and the final output may be smaller than the original data. Deflate (Deutsch, 1996) is an enhanced version of the Huffman coding that first uses a method called LZ77 (Ziv & Lempel, 1977) which replaces the repeat series of characters with the last known relative position within the sliding window and the lengths of the series of characters. The output of LZ77 is compressed again by Huffman coding. LZMA (Leavline & Singh, 2013) is another way of compression similar to Deflate but replaces the Huffman coding with probabilistic modeling-based range encoding. ZSTD (Collet & Kucherawy, 2021) also uses LZ77 first to compress the raw data and then uses Huffman coding to compress the literal characters outputted by LZ77; finally it uses a method called finite-state entropy to encode the rest of LZ77 outputs.

## 3 Proposed Method: eDQA

The goal of eDQA is to efficiently quantize DNN activations, providing high accuracy while minimizing computation and memory requirements, which is especially important in resource-constrained edge devices. Figure 1 gives an overview of eDQA. Given a target of quantization bits $N$:

- ① Offline, eDQA uses training/calibration data to rank the activation channels (i.e., determine their importance) using a greedy search algorithm that measures the impact of quantization on the target activations (see Section 3.1). Generally, the higher the accuracy required, the more important channels are needed (we can determine them using a tunable pre-selected ratio $r\%$ of the total channels in each layer) in the ranks.

- ② During inference, eDQA quantizes the important activation channels using $m$ extra bits (i.e., $N+m$ bits in total) while the rest of the channels are quantized using $N$ bits. Then eDQA right-shifts the important activation channels by $m$ bits and saves the shifting errors. For non-important channels, eDQA uses the direct quantization method;

- ③ During inference, eDQA compresses the shifting errors for important channels (see Section 3.2), which reduces the memory requirement.

- ④ During inference, eDQA de-quantizates the activation channels. For important channels, it decompresses shifting errors and adds them to the quantized activation channel values, thus compensating the information loss. For non-important channels, it uses the direct method to de-quantize.

Algorithms 1 and 2 describe steps ②, ③ and ④ of eDQA in more detail for a single layer. Note that channels are quantized/de-quantized one by one (see line 5 and line 4 in Algorithms 1 and 2 respectively); *se* in Algorithm 1 refers to the shifting error. Also note that $I$ are the important channel IDs. Finally, the shifting errors are obtained by reading the lower $m$ bits of each activation value before shifting. Since there are $2^m$ possible combinations of lower $m$ bits that correspond to the shifting errors, they can be converted to decimal floating point values using a pre-computed table (which maps the lower $m$ bits to shifting errors).

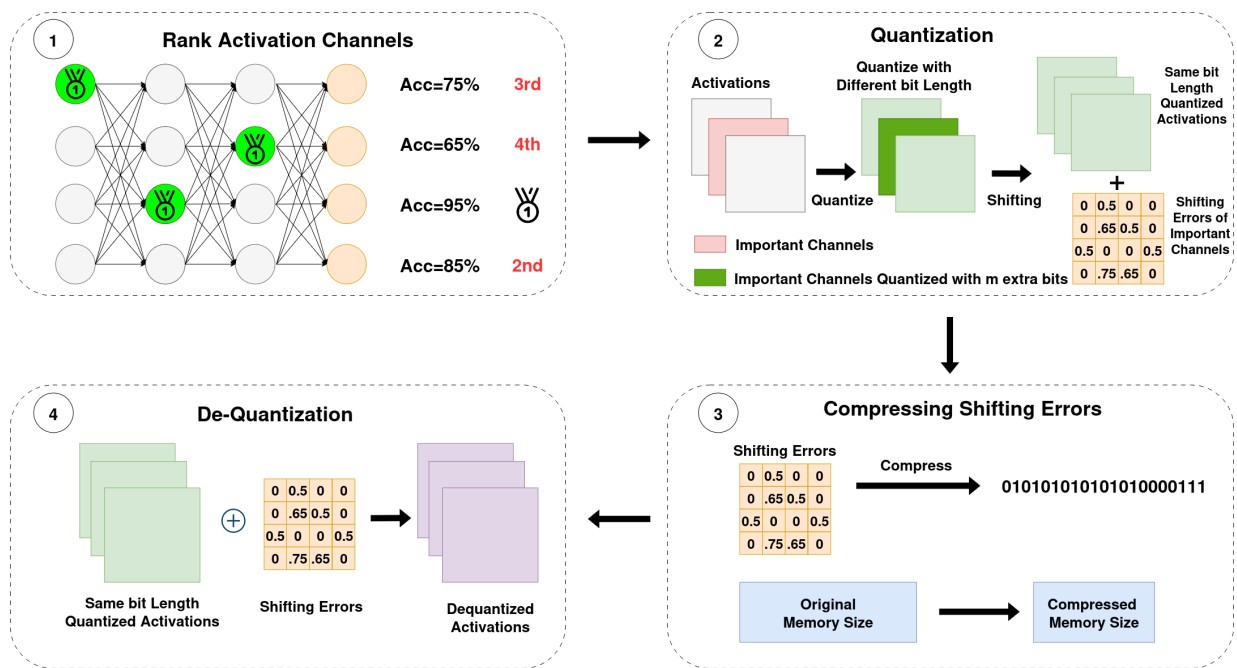

Figure 1: eDQA overview. ① offline, rank the activation channels based on importance using training/calibration data and a greedy search algorithm (green circles represent the most important channels for which we skip quantization); ② during inference, quantize important activation channels with $m$ extra bits and then right-shift them while saving the shifting errors; ③ the shifting errors are compressed to reduce the memory requirement; ④ de-quantize activation channels. For important channels, decompress shifting errors and add them to the quantized activation channel values. For non-important channels, use the direct method to de-quantize.

---

**Algorithm 1** eDQA Layer Quantization

---

1: **Input:** Activation Channels $A$, Target bits $N$, Extra bits $m$, Important channel ratio $r$, Channel Rank $R$, Data Compressor C(.)
2: **Output:** Quantized $A_q$, Encoded Shifting Errors $e$
3: $e = empty$
4: $I = $ get top $r\%$ channel IDs from $R$
5: **for** $a \in A$ **do**
6:     **if** $a.ID \in I$ **then**
7:         $\Delta_{N+m} = \frac{|max(A_{layer})|}{2^{N+m-1}}$
8:         $a_{more} = \text{Round}(\frac{a}{\Delta_{N+m}})$
9:         $a, se = \text{RightShift}(a_{more}, m)$
10:         $e = e \cup \text{C}(se)$
11:     **else**
12:         $\Delta_N = \frac{|max(A_{layer})|}{2^{N-1}}$
13:         $a = \text{Round}(\frac{a}{\Delta_N})$
14: $A_q = A$
15: **return** $A_q, e$

---

## 3.1 Ranking Important Activation Channels

We hypothesize that there are important activation channels that are best treated differently during quantization, similar to how AWQ (Lin et al., 2024) processes weights. We define important activation channels as those for which skipping quantization yields better prediction accuracy for the DNN. Since activations are dynamically generated, it would be computationally expensive to rank activation channels during inference.

---

**Algorithm 2** eDQA Layer De-Quantization

---
1: **Input:** Quantized Activation Channels $A_q$, Target bits $N$, Encoded Shifting Errors $e$, Important Channel Ratio $r$, Channel Rank $R$, Data Decompressor D(.)
2: **Output:** De-Quantized $A_{deq}$
3: $I$ = get top $r\%$ channel Ids from $R$
4: **for** $a \in A_q$ **do**
5:     $\Delta_N = \frac{|max(A_{layer})|}{2^{N-1}}$
6:     **if** $a.ID \in I$ **then**
7:         $a = \Delta_N \cdot (a + D(e))$
8:     **else**
9:         $a = \Delta_N \cdot a$
10: $A_{deq} = A_q$
11: **return** $A$

---

We therefore compute the ranks offline using training/calibration data. Assuming that both training/calibration and inference data are from similar distributions (which means they are processed similarly by the same DNN model), the ranks computed offline can be reused during inference. Therefore, we select fixed important channels based on a tunable preselected ratio $r\%$ of the total channels in each layer offline.

To calculate the rank of the activation channels, we avoid brute force and dynamic programming approaches due to their impractical time consumption. Instead, we use greedy search that only requires $O(LC)$ operations, where $C$ is the maximum number of activation channels of a layer, and $L$ is the number of layers. In our greedy search algorithm (see Algorithm 3), we iterate the activation channels for each layer. In each iteration, we skip the current activation channel and only quantize the remaining activation channels of the layer. We then run inference on the evaluation data to measure the impact of skipping the current activation channel on accuracy. The skipped activation channels that provide the highest accuracy are considered the most important ones for that layer. From layer two onward, to account for the impact from the previous layers (since an activation channel of a layer is related to the activation channels from the previous layers), we first quantize the activation channels of the previous layers except the most important channel of each layer that has already been determined (greedy selection).

---

**Algorithm 3** Ranking Important Activation Channels

---
1: **Input:** Model $M$, Training or Calibration Dataset $D$
2: **Output:** All ranks of activation channels $R$
3: $R = \{\}$
4: $P = \{\}$     // Most important activation channels of previous layers
5: **for** $layer \in M$ **do**     // In forward direction
6:     $rank = \{\}$
7:     $highest\_accuracy = 0$
8:     $most\_important\_channel = none$
9:     **for** $channel \in layer.activation\_channels$ **do**
10:         // IQ(.) quantizes activations of previous layers while skipping channels in $P$, and quantizes activations of current layer while skipping current channel. Then, IQ(.) gets inference accuracy.
11:         $accuracy = IQ(M, layer, channel, P, D)$
12:         $rank = rank \cup (channel, accuracy)$
13:         **if** $accuracy > highest\_accuracy$ **then**
14:             $highest\_accuracy = accuracy$
15:             $most\_important\_channel = channel$
16:     $P[layer] = most\_important\_channel$
17:     $rank = sort(rank)$     // sort by accuracy
18:     $R[layer] = rank$
19: **return** $R$

---

### 3.2 Quantizing Important Activation Channels

To preserve more information for the important activation channels (i.e., the top $r$ percent of all channels), and thus obtain higher accuracy, we quantize them with $m$ extra bits, where $m$ is less than or equal to the target number of bits $N$, and they are quantized using $N + m$ bits. Then we right-shift the activation values of the important channels by $m$ bits to reach the target bit length $N$. Note that this operation allows devices to store all quantized values with the same bit length $N$, which helps avoid wasting storage (Liu et al., 2024) or bus bandwidth. Also note that right-shifting is a computationally cheap operation (Elhoushi et al., 2021). Since right-shifting will lead to information loss for the important activation values, we save the shifting errors and add them back during the de-quantization phase to compensate the information loss.

#### 3.2.1 Quantization Error

We now formally analyze the shifting errors for the important activation channels. We denote the important activation channels by $I$. The most straightforward quantization/de-quantization approach for the important activation channels can be expressed as in AWQ (Lin et al., 2024):

$$\mathrm{Q}(I) = \Delta_N \cdot \mathrm{Round}(\frac{I}{\Delta_N}), \quad where \quad \Delta_N = \frac{|\max(I)|}{2^{N-1}} \tag{1}$$

The quantization error of this method is ($re$ is the Rounding Error):

$$Error = |I - \mathrm{Q}(I)| = |I - \Delta_N \cdot (\frac{I}{\Delta_N} \pm re)| = \Delta_N \cdot re \tag{2}$$

We hypothesize that the average $re$ is 0.25, which is similar to AWQ (Lin et al., 2024) for weight quantization. With eDQA, $I$ will be quantized with $m$ extra bits first, and then right-shifted also with $m$ bits. Then, the de-quantized important activation channels can be expressed as ($se$ is the Shifting Error):

$$\mathrm{Q}(I) = \Delta_N \cdot (\mathrm{RightShift}(\mathrm{Round}(\frac{I}{\Delta_{N+m}})) + se) \tag{3}$$

Finally, the quantization error becomes:

$$Error = |I - (\frac{\Delta_N}{2^m} \cdot (\frac{I}{\Delta_{N+m}} \pm re) - se + se)| = \Delta_N \cdot \frac{1}{2^m} \cdot re \tag{4}$$

From equations 2 and 4, it is clear that this approach exponentially ($2^m$) reduces the quantization error. This is expected since we add $m$ more bits of information. Note that the average $re$ is the sole source of error relative to the original arithmetic.

#### 3.2.2 Shifting Error Compression

We use data compression to compress shifting errors in order to reduce their storage overhead.

We first explore the shifting error values to justify their compressibility. We apply the operations from line 7 to 9 of Algorithm 1 to ResNet-32 on CIFAR-10, and collect the shifting errors and their frequencies. As we can see in Figure 2, the distributions of shifting errors are not uniform. Therefore, we can take advantage of this non-uniformity by applying compression techniques such as Huffman coding to exploit those patterns to save storage space.

We evaluate four different compression techniques, Huffman coding, Deflate, LZMA, and ZSTD, on two models (ResNet-32 and MobileNetV2) with CIFAR-10 (batch size is 128) and compared their compression ratio and latency. Note that the latency is the model inference time (averaged in batches) for eDQA with each compression technique running on one GPU (NVIDIA RTX 3090, as in Section 4) and one CPU (compression

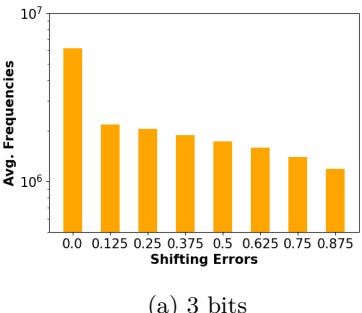
(a) 3 bits

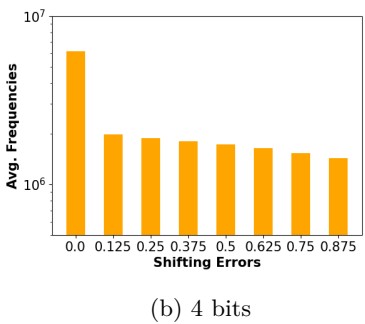
(b) 4 bits

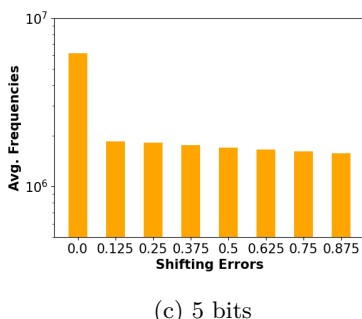
(c) 5 bits

Figure 2: Average frequency of shifting errors of ResNet-32 on CIFAR-10 with 3, 4, and 5 bits quantization with $m = 3$. Note, the clamp operations in quantization are skipped here, hence extreme values are preserved.

Table 1: Comparisons of different compression techniques used with eDQA on CIFAR-10 dataset with quantization level equals to 3 bits. The Latency is average eDQA inference times with the corresponding compression techniques per batch.

| Model | Compression Method | Compression Ratio | Latency (s) |
|---|---|---|---|
| ResNet-32 | Huffman | 1.12 | 3.47 |
| | Deflate | 1.12 | 4.62 |
| | LZMA | 1.12 | 8.4 |
| | ZSTD | 1.14 | 4.72 |
| MobileNetV2 | Huffman | 1.05 | 3.67 |
| | Deflate | 1.06 | 4.73 |
| | LZMA | 1.04 | 14.27 |
| | ZSTD | 1.06 | 4.78 |

and decompression is executed on CPU). As we can observe from Table 1, ZSTD provides the highest compression ratio (up to 1.14) while Huffman coding provides the smallest latency (up to 3.47s per batch on average). As a result, we suggest using Huffman coding while inference latency is critical, and ZSTD when compression ratio is important.

Note that, although the shifting errors can be compressed, they should not be included in the main memory (e.g., off-chip RAM) footprint or directly added when comparing memory footprints with prior methods, as they can instead be stored in on-chip memory.

## 4 Evaluation

### 4.1 Experimental Setup

We evaluate eDQA using ResNet-32 (He et al., 2016) and MobileNetV2 (Howard et al., 2017) on the CIFAR-10 dataset (Krizhevsky, 2009), and ResNet-18 and Vision Transformer (Dosovitskiy et al., 2021) on the TinyImageNet (Le & Yang, 2015) dataset.

We implement our experiments using PyTorch (Paszke et al., 2019) and run them on an NVIDIA RTX 3090 GPU, as the main goal of this section is to evaluate eDQA at the algorithm level (see Section 4.4 for validation on edge hardware). Every experiment is run 5 times if it involves randomness (for example, if it requires data sampling), and we take the average value in each case.

For each run of experiments of each DNN model, we created a rank table that contains the ranks of the activation channels (represented by their IDs) of each layer. Every rank is created with a random subset of the training data using greedy search (see Section 3.1). The size of the random training data subset for creating each rank map is 5000 for CIFAR-10 and 2500 for TinyImageNet. For EasyQuant, we use subsets

of the training data for calibration. The subset size is 5000 for all experiments except the TinyImageNet experiments, where we use a subset size of 2500. The batch size is 128 for all experiments. Note that all experiments only quantize activations, not weights, since this paper focuses only on activations. In this way, we can clearly show the effect of our method and exclude the impact of weight quantization.

## 4.2   Comparison of Different Methods

We compare eDQA with three methods: direct quantization (defined in Section 2), PoT quantization (with 1 bit for the sign of value, 1 bit for the sign of exponent, and the remaining bits for exponent), EasyQuant (Wu et al., 2020), and the state-of-the-art method NoisyQuant (Liu et al., 2023). Even though NoisyQuant was designed for Vision Transformers (Dosovitskiy et al., 2021), our experiments show that it also works well for other types of DNNs.

Since eDQA is designed for deep quantization, we set the target quantization levels to 3, 4, and 5 bits; note that $m = 3$ in all cases to simplify the evaluation. We empirically set the important channel ratio $r$ to 55% for ResNet-18 and 40% for all other models.

Table 2 shows the accuracy results for the three DNN models. In general, eDQA provides higher accuracy in most cases (12 out of 15). More specifically, eDQA achieves up to 86.63 percentage points higher accuracy than the compared methods (eDQA improves accuracy by 86.63 percentage points over EasyQuant in the 5-bit quantization experiment on ViT-B/16 with TinyImageNet.). For ResNet-18 at 3 bits quantization, we notice eDQA accuracy is lower than NoisyQuant; we attribute this to the relatively smaller size of this model. With a smaller model size, most of the activation channels are important to ResNet-18, which means that the higher precision provided by eDQA may be less substantial. However, as the bit length increases, eDQA starts to beat all the comparisons for that case. We also observe that the lower the quantization bits, the more accuracy advantages eDQA can provide. This is because the shorter bit lengths contain less information, but eDQA mitigates this by using extra $m$ bits to quantize the important activation channels.

## 4.3   Impact of Hyperparameters

There are two important hyperparameters in eDQA, the extra bits $m$ and the important channel ratio $r$. In order to understand how they impact the accuracy of eDQA, we evaluate them by fixing one of them and tuning the other.

First, we evaluate the accuracy of two models (ResNet-32 and MobileNetV2) on CIFAR-10 using eDQA with different important channel ratios $r$, from 10% to 90% (every 10 %) by fixing $m = 3$ and setting the quantization level to 3 bits. Note that $r = 0\%$ is equivalent to the direct method in Table 2. As we observe from Figure 3, the higher the important ratios, the higher the accuracy is. This is because the more channels get extra bits, the more information is preserved. However, the gain in accuracy slows down after certain important channel ratios; in Figure 3 we find that it is 50% for both models. That shows that the impact of important ratios on accuracy is reduced. We believe that it is because with increasing important channel ratios, the more unimportant channels get extra bits, but they cannot provide significant important information even with extra bits.

Second, we evaluate the accuracy of the same two models (ResNet-32 and MobileNetV2) on CIFAR-10 using eDQA with different numbers of extra bits $m \in \{1, 2, 3\}$, fixing $r = 40\%$, and setting the quantization level to 3 bits. As we observe in Figure 4, the more extra bits $m$, the higher the accuracy. This is because more extra bits help to preserve more information. However, the improvement in accuracy from $m = 2$ to $m = 3$ is significantly smaller than the improvement from $m = 1$ to $m = 2$. This suggests that the value of $m$ does not need to be maximized, as the marginal benefit of further increases is minimal.

## 4.4   Edge Device Evaluation

To evaluate the feasibility and effectiveness of eDQA on edge hardware platforms, we design, deploy, and benchmark a proof-of-concept (PoC) hardware accelerator that incorporates eDQA quantization and de-quantization on an edge FPGA. While this paper mainly focuses on the algorithmic design of eDQA, this

Table 2: Accuracy results for the three DNN models under study. 3 Bits, 4 Bits, 5 Bits means the accuracy get with 3, 4, 5 bits quantization.

| Model, Dataset, Accuracy (%) | Method | 3 Bits (%) | 4 Bits (%) | 5 Bits (%) |
|---|---|---|---|---|
| ResNet-32, CIFAR-10, Ori Acc: 92.49 | Direct | 52.85 | 87.74 | 91.33 |
| | PoT | 28.17 | 79.90 | 80.22 |
| | EasyQuant | 49.03 | 87.84 | 91.79 |
| | NoisyQuant | 71.22 | 89.66 | 91.86 |
| | eDQA(m = 3) | **82.13** | **90.55** | **92.12** |
| MobileNetV2, CIFAR-10, Ori Acc: 91.42 | Direct | 82.75 | 90.01 | 91.19 |
| | PoT | 13.63 | 71.18 | 89.68 |
| | EasyQuant | 77.61 | 90.44 | 91.08 |
| | NoisyQuant | 87.67 | 90.56 | **91.33** |
| | eDQA(m = 3) | **88.63** | **90.84** | 91.28 |
| ViT-b-16, TinyImageNet, Ori Acc: 88.31 | Direct | 15.07 | 79.69 | 86.37 |
| | PoT | 29.54 | 81.57 | 82.69 |
| | EasyQuant | 0.53 | 0.58 | 0.68 |
| | NoisyQuant | 60.36 | 83.05 | 85.75 |
| | eDQA(m = 3) | **63.41** | **84.52** | **87.31** |
| ResNet-18, TinyImageNet, Ori Acc: 72.51 | Direct | 49.06 | 67.53 | 71.66 |
| | PoT | 8.58 | 67.34 | 67.66 |
| | EasyQuant | 10.59 | 62.22 | 71.04 |
| | NoisyQuant | **65.05** | 70.53 | 71.57 |
| | eDQA(m = 3) | 63.61 | **70.63** | **71.89** |
| ResNet-18, ImageNet, Ori Acc: 67.29 | Direct | 33.16 | 63.32 | 66.60 |
| | PoT | 17.83 | 63.91 | 63.97 |
| | EasyQuant | 2.40 | 53.24 | 65.98 |
| | NoisyQuant | **60.22** | 65.36 | 66.76 |
| | eDQA(m = 3) | 53.86 | **65.75** | **67.05** |

FPGA experiment validates its implementation in a more realistic setting. For this implementation, we use Huffman compression in eDQA, as this provides the best latency in our initial evaluation, as seen in Table 1. We note that, while variable-length Huffman encoding requires random memory access, it may increase memory access latency, particularly on general-purpose hardware. By leveraging the ability of FPGAs to implement customized memory hierarchies, potential bottlenecks introduced by variable-length Huffman encoding can be alleviated. We do not further investigate Huffman compression optimization on custom hardware, as it is beyond the scope of this work.

### 4.4.1 Accelerator Design

Using the SECDA design methodology (Haris et al., 2021), we design and develop a custom FPGA-based hardware accelerator for eDQA to measure its performance on a resource-constrained edge device. We use the PYNQ-Z1 (Digilent, 2020) board, that includes an edge FPGA with limited on-chip memory and compute capacity, as our target device; this device also contains a dual-core ARM A9 processor and 512 MB of memory.

Our eDQA accelerator, which was designed to process a single input channel at a time. Figure 5 provides a visualization of the accelerator design with its key four components: *Quantizer*, *Huffman Encoder*, *Huffman Decoder*, and *DeQuantizer*. First, the *Quantizer* performs the quantization step, storing the N-bit 'quantized_data'. Second, the *Huffman Encoder* is used to create a Huffman tree for the 'shifting_error' values. This Huffman tree is then used to generate the codebook that is used to compress the 'shifting_error' into the 'compressed_error'. The codebook is then encoded and stored in on-chip memory alongside the compressed 'compressed_error'. To de-quantize, *Huffman Decoder* is used to recreate the Huffman tree using the 'encoded

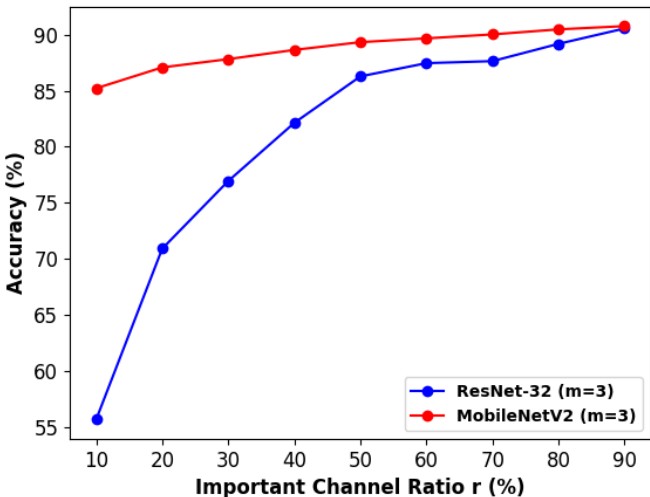

Figure 3: The impact of different $r$ values and $m = 3$. Quantization level is 3 bits.

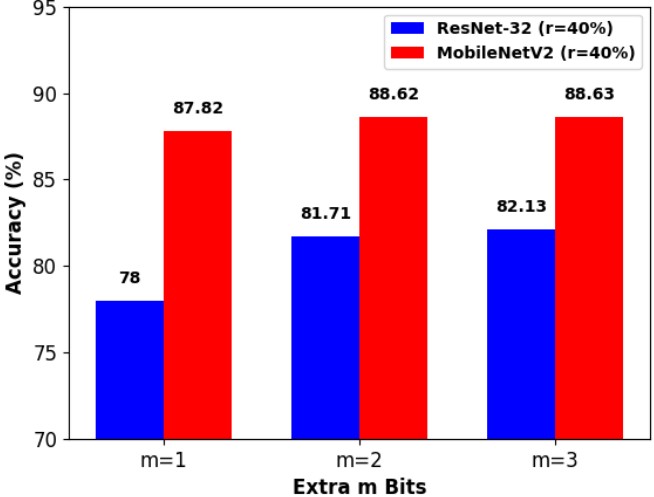

Figure 4: The impact of different $m$ values and $r = 40\%$. Quantization level is 3 bits.

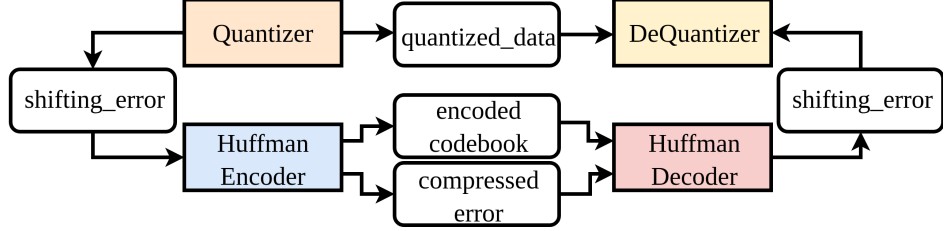

Figure 5: Overview of the eDQA accelerator.

codebook', which is then used to decompress the 'compressed_error' into the original 'shifting_error'. Finally, the *DeQuantizer* processes 'quantized_data' and 'shifting_error' to calculate the de-quantized data.

### 4.4.2 Hardware Results

To evaluate and compare the effectiveness of our eDQA accelerator, we also implemented a hardware accelerator for the NoisyQuant (Liu et al., 2023) method targeting the same FPGA device. Note that for this experiment, we used random input data that have the same size as the target layers. For our eDQA accelerator, we only consider the acceleration of important channels and we set $N = 3$ and $m = 3$ for 'quantized_data' and 'shifting_error', respectively. NoisyQuant requires some additional configurations, which we set as follows: i) the forward pass function, required for NoisyQuant's quantization, as Matrix Multiplication; ii) we use the input data as weights for the forward pass function; iii) the activation scale search iterations as 10; iv) the noise search iterations as 10; v) the bit-width as 3.

We only evaluate per-channel quantization/de-quantization performance to understand the performance per channel. In a full accelerator design, the hardware designer would deploy multiple parallel instances of our eDQA hardware accelerator to satisfy the performance needs of the accelerator design. Table 3 shows the performance of our eDQA accelerator compared to the NoisyQuant implementation in terms of latency across the DNN layers of ResNet-18. We achieve a speedup of up to $309\times$ on the largest problem size; we attribute this to the algorithmic simplicity of eDQA compared to the complex online search used by NoisyQuant.

Table 3: Latency comparison between eDQA and NoisyQuant methods for ResNet-18 layers. Size denotes the input data size in 1D. Latencies are shown in microseconds (Quant/DeQuant).

| Layer | Size | eDQA [µs] | Noisy [µs] | Speedup |
|-------|------|-----------|------------|---------|
| 7 & 8 | 49 | 24 / 4 | 1488 / 7 | 53× |
| 5 & 6 | 196 | 85 / 14 | 9231 / 28 | 93× |
| 3 & 4 | 784 | 329 / 52 | 63395 / 114 | 167× |
| 1 & 2 | 3136 | 1303 / 205 | 465593 / 455 | 309× |

Table 4: Hardware resource utilization (and utilization percentage) comparison between our eDQA and NoisyQuant hardware implementations.

| Hardware | BRAM | DSP | FFs | LUTs |
|----------|------|-----|-----|------|
| eDQA | 42 (15%) | 12 (5%) | 14.6K (13%) | 16.8K (31%) |
| NoisyQuant | 71 (25%) | 40 (18%) | 21.6K (20%) | 24.7K (46%) |

Finally, Table 4 shows the resource utilization of our eDQA accelerator compared to the NoisyQuant implementation. As evidenced by its low resource utilization, our eDQA accelerator is very lightweight and can easily be scaled to improve parallelism to process multiple channels simultaneously.

## 5 Conclusion

We proposed eDQA, an efficient method that applies deep quantization (with less than 6 bits) to DNN activations and provides high accuracy while being suitable for resource-constrained edge devices. We evaluated eDQA on four different models and dataset combinations, showing up to 75% accuracy improvement compared to existing methods. Additionally, and as a proof of concept, we implement and evaluate eDQA on an edge FPGA, achieving up to $309\times$ speedup compared to a NoisyQuant implementation on the same device. This work has several limitations. First, we assume that the offline calibration dataset follows the same distribution as the online data, which does not account for the impact of out-of-distribution online data on identifying important activation channels. Second, eDQA introduces additional activation operations and hardware overheads, such as extra energy costs from compression operations, which warrant further investigation. Third, we only provide a PoC accelerator design, leaving a complete hardware–algorithm co-design as an avenue for future exploration. As future work, we plan to systematically investigate the impact of out-of-distribution online data, quantify the hardware-related overheads of eDQA, further improve its memory efficiency, and co-design hardware accelerators (Gibson et al., 2025) for end-to-end evaluation.

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
