# OpenReview forum: "eDQA: Efficient Deep Quantization of DNN Activations on Edge Devices"
_TMLR — Decision pending for TMLR_

### Review · Reviewer_xiQx · 2025-12-06

**Summary Of Contributions:**

The paper proposes eDQA, an efficient deep quantization framework designed to achieve sub-6-bit activation quantization on resource-constrained edge devices without relying on expensive online hyperparameter searches. This method identifies important activation channels offline and preserves their precision using a novel shifting-based technique combined with error compression, allowing the system to maintain hardware-friendly uniform bit-lengths while effectively utilizing mixed-precision principles. Experimental results demonstrate that eDQA significantly outperforms state-of-the-art methods like NoisyQuant, achieving up to 75% higher accuracy and delivering up to a 309× speedup on FPGA hardware.

**Audience:**

Yes

**Audience Explanation:**

Model quantization is a crucial field, particularly when deploying models to resource-constrained environments. It involves **reducing the numerical precision of model parameters** (for example, converting from 32-bit floating-point to 8-bit integers), which **significantly reduces memory usage, increases computational speed, and lowers power consumption** with minimal loss of accuracy. This not only enables large models to run efficiently on mobile phones, embedded systems, and edge computing devices but also substantially cuts deployment and serving costs in cloud environments. It is a key step towards the large-scale practical application of AI technology.

**Broader Impact Concerns:**

None. The paper focuses on algorithmic and hardware optimizations for neural network inference and does not raise specific ethical concerns.

**Claims And Evidence:**

Yes

**Claims Explanation:**

The authors substantiate their accuracy claims through extensive experiments on multiple architectures and datasets, demonstrating consistent improvements over state-of-the-art baselines like NoisyQuant. Furthermore, the efficiency and feasibility of the method are convincingly validated by a real-world FPGA implementation that empirically confirms the reported significant speedups and low hardware resource utilization.

**Requested Changes:**

1. The evaluation relies on small-scale datasets like CIFAR-10 and TinyImageNet, which are insufficient to demonstrate robustness for modern computer vision tasks. Standard quantization research requires validation on the full ImageNet dataset to prove the method works on complex real-world distributions.

2. The baseline comparison excludes widely adopted and effective methods such as LSQ, PACT, or SmoothQuant. Limiting the comparison to Direct, PoT, and NoisyQuant fails to establish the proposed method's superiority against the actual state-of-the-art.

3. The study exclusively focuses on activation quantization while leaving weight quantization unaddressed or in higher precision. This presents an incomplete solution for edge devices, where simultaneous quantization of both weights and activations is necessary for meaningful resource savings.

4. Using variable-length Huffman coding for shifting errors destroys random access capability in memory, which is critical for efficient hardware processing. This design choice introduces complex control logic and buffering requirements that negate the claimed memory efficiency benefits.


5. The reliance on an offline greedy search algorithm for channel ranking assumes that inference data strictly follows the calibration distribution. This static approach is brittle in dynamic edge environments where input statistics may drift or vary significantly from the calibration set.

6. The experiments primarily test lightweight models like ResNet-32 and MobileNetV2, failing to verify the method's scalability to larger, over-parameterized architectures. The paper does not provide evidence that the accuracy gains persist when the method is applied to deeper networks with higher channel redundancy.

---

### Review · Reviewer_C3ZN · 2025-12-19

**Summary Of Contributions:**

This paper introduces eDQA (Efficient Deep Quantization of DNN Activations), a novel activation quantization framework specifically designed for sub-6-bit inference on resource-constrained edge devices. The key idea is to treat activation channels heterogeneously based on their importance, while still preserving uniform final bit-widths for hardware efficiency.

The main contributions can be summarized as follows:

Channel-aware activation quantization: The authors propose an offline greedy ranking algorithm that identifies important activation channels based on their impact on inference accuracy. These channels are treated differently during quantization, allowing more information to be preserved where it matters most.

Shift-based mixed-precision quantization with uniform storage: Important channels are quantized using N+m bits and then right-shifted back to N bits, storing the lost lower-bit information as shifting errors. This enables effective mixed-precision quantization without introducing non-uniform bit-widths, which are typically inefficient for hardware.

Compression of shifting errors: The paper observes that shifting errors follow a non-uniform distribution and can therefore be efficiently compressed using standard compression techniques (Huffman, Deflate, LZMA, ZSTD), significantly reducing memory overhead.

Extensive empirical evaluation: eDQA is evaluated on four models (ResNet-32, MobileNetV2, ResNet-18, ViT-B/16) and two datasets (CIFAR-10, TinyImageNet), showing up to 75% accuracy improvement over direct quantization, PoT quantization, and the state-of-the-art NoisyQuant method for sub-6-bit activations.

Hardware validation on FPGA: The authors implement eDQA on an edge FPGA (PYNQ-Z1) and demonstrate up to 309× speedup compared to NoisyQuant, along with lower hardware resource utilization.

Strengths include a strong hardware-aware design, substantial accuracy improvements in very low-bit regimes, and convincing FPGA validation. Weaknesses mainly relate to assumptions about data distribution stability and the offline nature of channel ranking.

**Additional Comments:**

1) The paper is well written, clearly structured, and easy to follow. 2) Figures and algorithms are informative and well integrated into the text. 3) The combination of algorithmic innovation with hardware validation is a major strength. 4) The FPGA results are particularly compelling and help distinguish this work from purely software-level quantization methods.

**Audience:**

Yes

**Audience Explanation:**

The paper is highly relevant to several segments of the TMLR audience, including researchers and practitioners working on:

1) Efficient inference and model compression 2) Quantization-aware deep learning 3) Edge AI and hardware-aware ML system design
4) FPGA-based and low-power neural accelerators

The focus on activation quantization below 6 bits, combined with explicit hardware validation, addresses a gap in the literature where many prior works emphasize either algorithmic novelty without hardware feasibility or hardware efficiency without strong accuracy guarantees. The results and methodology are therefore likely to be of broad interest within the TMLR community.

**Broader Impact Concerns:**

The work focuses on improving the efficiency of DNN inference on edge devices. While there are no direct ethical concerns, enabling more powerful inference on low-cost or pervasive devices could have indirect implications related to privacy, surveillance, or misuse. The paper would benefit from a brief broader impact statement acknowledging that improved edge inference capabilities should be deployed responsibly, particularly in applications involving sensitive data or large-scale monitoring.

**Claims And Evidence:**

Yes

**Claims Explanation:**

The paper provides convincing evidence to support its claims through a combination of algorithmic design, empirical evaluation, and hardware implementation: The algorithmic motivation is clearly articulated: existing activation quantization methods either fail under sub-6-bit settings or rely on expensive computations unsuitable for edge devices.The greedy channel-ranking algorithm is well described and justified, with manageable computational complexity and practical offline execution. Quantization error analysis (Section 3.2) mathematically demonstrates that using m extra bits reduces quantization error exponentially by a factor of 2m, providing theoretical grounding for the approach. Accuracy results across multiple models and datasets consistently show strong gains over baselines, especially in the most challenging 3-bit setting. Hyperparameter studies (on the important channel ratio r and extra bits m) provide useful insights and show diminishing returns, indicating practical tunability.The FPGA evaluation strongly supports claims of efficiency and practicality, with dramatic speedups and lower resource utilization compared to NoisyQuant. Overall, the experimental methodology is sound, comparisons are fair, and conclusions are well supported by the presented results.

**Requested Changes:**

Critical (required for acceptance)

1) Clarify robustness of offline channel ranking:- The approach assumes that the activation distributions during inference closely match those in the training or calibration data used for ranking channels. A clearer discussion of how sensitive eDQA is to distribution shift (e.g., domain shift, input corruption) would strengthen the paper.

2) Generalization to other tasks and modalities:-  All experiments focus on image classification. A discussion (or small experiment) addressing whether the method generalizes to other tasks such as detection, segmentation, or non-vision workloads would improve the scope of the work.

3) Memory overhead analysis at system level:-  While shifting errors are compressed, a clearer end-to-end comparison of total memory footprint (quantized activations + compressed errors) versus baseline methods would help quantify the real system-level benefits.

Non-critical (would strengthen the paper)

1) Provide a clearer comparison with other recent activation quantization methods such as SmoothQuant-style approaches, even if only conceptually. 2) Include ablations on different values of  m for higher-bit targets (e.g., 4-bit or 5-bit) to see whether trends remain consistent. 3) Add a short discussion on how eDQA could be combined with weight quantization for full end-to-end compression.

---

### Review · Reviewer_79RK · 2026-01-25

**Summary Of Contributions:**

In this paper, the authors address the challenge of low-bit (sub-6-bit) activation quantization in DNNs. Existing methods like NoisyQuant are computationally prohibitive for edge devices due to expensive online parameter searching. Furthermore, while standard mixed-precision methods exist, they often result in wasted storage and bus bandwidth due to irregular bit-widths. To address this, the authors propose eDQA, which identifies important activation channels offline. These channels are quantized with extra m bits for higher precision and then right-shifted to match the target uniform bit-width. The resulting 'shifting errors' are compressed using techniques like Huffman coding. Validated on an FPGA, the method achieves up to a 309x speedup compared to NoisyQuant

**Audience:**

Yes

**Audience Explanation:**

Author implemented the algorithms on FPGA and the speed is 309x faster than NoisyQuant. This the engineer-wise significant improvement. So,Yes, this paper addresses the critical problem of efficient inference on resource-constrained edge devices

**Broader Impact Concerns:**

No specific ethical concerns. The paper focuses on low-level model quantization and hardware acceleration techniques to improve energy and computational efficiency on edge devices. It does not introduce new datasets or discriminatory algorithms

**Claims And Evidence:**

No

**Claims Explanation:**

Major claim is supported well, but the experiment could be more valid. The claim 'High Inference Speed on Edge Devices' is supported well. The claim ''"up to 75% better accuracy'' could be verified on more large-scale dataset to show the robustness. The result for 'avoid storage and bus bandwidth waste' is not so convincing

**Requested Changes:**

(-) I am interested in seeing the performance improvements compared to NoisyQuant on the full ImageNet dataset, as results on TinyImageNet may not fully demonstrate the method's scalability.

(-) while Figures 3 and 4 report performance gains, the associated storage overhead is not discussed. It is unclear whether the proposed 3-bit method, including its overhead, might actually result in a larger memory footprint than a standard 4-bit baseline. To demonstrate that eDQA is truly efficient, please provide a comparison against a standard Uniform 4-bit quantization baseline and include a plot of Accuracy vs. Actual Model Size (MB).

---

### Comment · Reviewer_C3ZN · 2026-02-19
**I cant see any response from Authors**

Want to confirm this again.

---

> ### Author Response · Authors · 2026-03-23
> **Responses for common questions**
>
> ## **Responses for common questions:**
> ### **C1. Larger Dataset: (Reviewers 79RK and xiQx)**
> **RC1:** Thanks for your comments on the evaluation of eDQA on a larger dataset like ImageNet. To address your concerns, we evaluated eDQA with ResNet-18 on ImageNet.
>
> **Table 1: Method comparison for ResNet-18 on ImageNet.**
> | Method | 3-Bit Acc (%)| 4-Bit Acc (%)| 5-Bit Acc (%) |
> |----------|----------|----------|----------|
> | Baseline   |33.16   | 63.32   | 66.6   |
> | PoT  |  17.83  | 63.91   | 63.97  |
> | NoisyQuant | **60.22**  | 65.36  |  66.76  |
> | eDQA | 53.86  | **65.75** | **67.05**  |
>
> As shown in Table 1, eDQA is the winner in most cases (2 out of 3). This demonstrates that eDQA has advantages with the ImageNet dataset. We will include the new results in the final version of our paper.
> ### **C2. Actual model size and overheads: (Reviewers 79RK and C3ZN)**
> **RC2:** We thank the reviewers for raising concerns about the overhead of eDQA. Our design is hardware-aware: the overhead is compressed and stored separately from quantized activations, and can be placed in on-chip cache or other high-efficiency storage rather than main memory. Additionally, the overhead size is not fixed and depends on the proportion of channels designated as important, which can be tuned to balance accuracy and inference speed. Therefore, the overhead should not be directly counted as part of main memory (e.g., RAM) usage, nor simply added when comparing memory footprints with prior methods. We will clarify this point in the final version of the paper.
> ### **C3. Weight quantization: (Reviewers C3ZN and xiQx)**
> **RC3:** We thank the reviewers for their comments on weight quantization. In this work, we focus exclusively on activation quantization and assume that weights can be quantized using any suitable existing method. Unlike approaches that jointly optimize weights and activations, our method is independent of weight quantization, which provides greater flexibility in practice, and we will clarify this design choice and include a corresponding discussion in the final version of the paper.
> ### **C4. Robustness of offline channel ranking of eDQA: (Reviewers C3ZN and xiQx)**
> **RC4:** We thank the reviewers for their suggestion regarding the robustness of eDQA. Our approach is inspired by AWQ [1], which uses a calibration dataset to determine the hyperparameter. Similarly, the channel ranking in eDQA can be viewed as a hyperparameter and can therefore be derived using a calibration dataset. We will clarify this point and add a discussion on the robustness of the offline channel ranking in the final version of the paper.
> ### **C5. Compare with other methods: (Reviewers C3ZN and xiQx)**
> **RC5:** We thank the reviewers for their suggestions on comparing eDQA with additional popular methods, including LSQ [2], PACT [3], and SmoothQuant [4]. However, SmoothQuant requires joint optimization of weights and activations, while LSQ and PACT are quantization-aware training (QAT) methods. In contrast, eDQA is a post-training method that focuses solely on activation quantization and operates independently of weight quantization, making direct comparisons less appropriate. Nevertheless, to broaden the empirical evaluation, we include comparisons with EasyQuant [5], a representative activation-focused baseline also used in the NoisyQuant paper.
>
> **Table 2: Comparisons of eDQA and EasyQuant (EQ)**
> | Method| Model|Dataset| 3-Bit Acc (%)| 4-Bit Acc (%)| 5-Bit Acc (%)|
> |----------|----------|----------|----------|----------|----------|
> |EQ| ResNet-32| CIFAR-10 |49.03 | 87.84 | 91.79|
> |eDQA| ResNet-32| CIFAR-10 |**82.13**| **90.55**| **92.12** |
> |EQ|MobileNetV2  |CIFAR-10| 77.61  | 90.44 | 91.08|
> |eDQA|MobileNetV2  |CIFAR-10| **88.63**| **90.84**| **91.28** |
> |EQ|ViT-B/16|Tiny-ImageNet | 0.53 | 0.58 | 0.68 |
> |eDQA|ViT-B/16|Tiny-ImageNet|**63.41**|**84.52**|**87.31**|
> |EQ|ResNet-18 |Tiny-ImageNet | 10.59 | 62.22 | 71.04|
> |eDQA|ResNet-18 |Tiny-ImageNet|**63.61**|**70.63**|**71.89**|
>
> As shown in Tables 2, eDQA consistently outperforms EasyQuant, achieving improvements of up to 86.63%. These results highlight the strong potential of eDQA to deliver better accuracy than existing activation quantization methods. We will include these results in the final version of the paper.
> ### **C6. Hardware issues: (Reviewer 79RK and xiQx)**
> **RC6:** We thank the reviewers for their comments on the hardware efficiency of eDQA. Our method enables the use of lower-precision data buses in custom accelerator designs, allowing for reduced energy consumption or, alternatively, increased bandwidth by allocating additional hardware resources (e.g., LUTs). While variable-length Huffman encoding is less suited to general-purpose hardware, custom FPGA-based accelerators with specialized memory hierarchies can support low-latency random access and efficient decoding. We will clarify these points and add a corresponding discussion in the final version of the paper.

---

> ### Author Response · Authors · 2026-03-23
> **Responses to individual questions for reviewer 79RK**
>
> **Reviewer 79RK:**
>
> **Q11:** Major claim is supported well, but the experiment could be more valid. The claim 'High Inference Speed on Edge Devices' is supported well. The claim ''"up to 75% better accuracy'' could be verified on more large-scale dataset to show the robustness.
>
> **R11:** Addressed in RC1.
>
> ---
>
> **Q12:** The result for 'avoid storage and bus bandwidth waste' is not so convincing.
>
> **R12:** Addressed in RC6.
>
> ---
>
> **Q13:** I am interested in seeing the performance improvements compared to NoisyQuant on the full ImageNet dataset, as results on TinyImageNet may not fully demonstrate the method's scalability.
>
> **R13:** Addressed in RC1.
>
> ---
>
> **Q14:** While Figures 3 and 4 report performance gains, the associated storage overhead is not discussed. It is unclear whether the proposed 3-bit method, including its overhead, might actually result in a larger memory footprint than a standard 4-bit baseline. To demonstrate that eDQA is truly efficient, please provide a comparison against a standard Uniform 4-bit quantization baseline and include a plot of Accuracy vs. Actual Model Size (MB).
>
> **R14:** Addressed in RC2.

---

> ### Author Response · Authors · 2026-03-23
> **Responses to individual questions for reviewer C3ZN**
>
> **Reviewer C3ZN:**
>
> **Q21:** Clarify robustness of offline channel ranking. The approach assumes that the activation distributions during inference closely match those in the training or calibration data used for ranking channels. A clearer discussion of how sensitive eDQA is to distribution shift (e.g., domain shift, input corruption) would strengthen the paper.
>
> **R21:** Addressed in RC4.
>
> ---
>
> **Q22:** Generalization to other tasks and modalities. All experiments focus on image classification. A discussion (or small experiment) addressing whether the method generalizes to other tasks such as detection, segmentation, or non-vision workloads would improve the scope of the work.
>
> **R22:** We thank the reviewer for their comments on applying eDQA to other tasks. eDQA operates at the numerical level, making it inherently task-agnostic. Additionally, prior work [6] indicates that computer vision workloads dominate on edge devices, which supports our choice of evaluation tasks. We will clarify this point and include a corresponding discussion in the final version of the paper.
>
> ---
>
> **Q23:** Memory overhead analysis at system level. While shifting errors are compressed, a clearer end-to-end comparison of total memory footprint (quantized activations + compressed errors) versus baseline methods would help quantify the real system-level benefits.
>
> **R23:** Addressed in RC2.
>
> ---
>
> **Q24:** Provide a clearer comparison with other recent activation quantization methods such as SmoothQuant-style approaches, even if only conceptually.
>
> **R24:** Addressed in RC5.
>
> ---
>
> **Q25:** Include ablations on different values of m for higher-bit targets (e.g., 4-bit or 5-bit) to see whether trends remain consistent.
>
> **R25:** We thank the reviewer for the comments regarding higher-bit m. Our design primarily targets sub-6-bit activation quantization, with a focus on 3-, 4-, and 5-bit settings. Under these low-bit regimes, 4- or 5-bit values for m can be relatively large compared to the quantized activations, which may increase the associated overhead. We will clarify this design assumption and its implications in the final version of the paper.
>
> ---
>
> **Q26:** Add a short discussion on how eDQA could be combined with weight quantization for full end-to-end compression.
>
> **R26:** Addressed in RC3.

---

> ### Author Response · Authors · 2026-03-23
> **Responses to individual questions for reviewer xiQx**
>
> **Reviewer xiQx:**
>
> **Q31:** The evaluation relies on small-scale datasets like CIFAR-10 and TinyImageNet, which are insufficient to demonstrate robustness for modern computer vision tasks. Standard quantization research requires validation on the full ImageNet dataset to prove the method works on complex real-world distributions.
>
> **R31:** Addressed in RC1.
>
> ---
>
> **Q32:** The baseline comparison excludes widely adopted and effective methods such as LSQ, PACT, or SmoothQuant. Limiting the comparison to Direct, PoT, and NoisyQuant fails to establish the proposed method's superiority against the actual state-of-the-art.
>
> **R32:** Addressed in RC5.
>
> ---
>
> **Q33:** The study exclusively focuses on activation quantization while leaving weight quantization unaddressed or in higher precision. This presents an incomplete solution for edge devices, where simultaneous quantization of both weights and activations is necessary for meaningful resource savings.
>
> **R33:** Addressed in RC3.
>
> ---
>
> **Q34:** Using variable-length Huffman coding for shifting errors destroys random access capability in memory, which is critical for efficient hardware processing. This design choice introduces complex control logic and buffering requirements that negate the claimed memory efficiency benefits.
>
> **R34:** Addressed in RC6.
>
> ---
>
> **Q35:** The reliance on an offline greedy search algorithm for channel ranking assumes that inference data strictly follows the calibration distribution. This static approach is brittle in dynamic edge environments where input statistics may drift or vary significantly from the calibration set.
>
> **R35:** Addressed in RC4.
>
> ---
>
> **Q36:** The experiments primarily test lightweight models like ResNet-32 and MobileNetV2, failing to verify the method's scalability to larger, over-parameterized architectures. The paper does not provide evidence that the accuracy gains persist when the method is applied to deeper networks with higher channel redundancy.
>
> **R36:** We thank the reviewer for the suggestion regarding evaluation on large models. The primary motivation behind eDQA is efficient deployment on edge devices, where resource constraints typically necessitate the use of compact architectures such as ResNets or MobileNetV2. While applying eDQA to larger models is feasible, it is less aligned with the target scenario of edge deployment. We will clarify this motivation and include a discussion in the final version of the paper.

---

> > ### Comment · Reviewer_79RK · 2026-03-27
> > **Comment on Author's Rebuttal**
> >
> > Overall, the paper's core claims like significant accuracy improvement / ultra-fast edge inference / exponential quantization error reduction are well supported. however, after carefully reading the rebuttal and other reviewers' concerns, I have two remaining concerns.
> >
> > 1 . overclaim on hardware efficiency
> > The paper argues that existing mixed-precision methods are 'not hardware friendly due to storage and bus bandwidth waste (in related work)', implicitly suggesting that eDQA is a broadly hardware-efficient solution. However, eDQA relies on customized FPGAs, and the total overhead of these custom hardware requirements is never rigorously discussed, nor do the authors consider it necessary to discuss (as said in author's rebuttal). So I think the hardware efficiency claims should be more explicitly scoped, and the method's hardware limitations should be clearly stated. I wonder other reviewers' thought on that.
> >
> > 2. robustness to the domain shift
> > The authors use AWQ to argue that their offline channel ranking can be treated as a calibration-derived hyperparameter, and thus domain shift is not a fundamental concern. This analogy is misleading. AWQ explicitly claims robustness to domain shift as they say they can avoid overfitting to the calibration set. Crucially, AWQ's scaling is sort of soft operation. In contrast, eDQA's context is about hard bit-level truncation to channels deemed unimportant. Therefore, it is natureally that eDQA is more sensitive to data distribution shift than AWQ. AWQ's explicit robustness claim actually is highlighted in their paper. Can the authors provide further justification?
> >
> > I raise above concerns as important discussion points and questions for this discussion phase. Overall, I think major claims (even some of them is not explicitly scoped well) have good support.

---

> ### Author Response · Authors · 2026-03-23
> **References**
>
> **References:**
>
> [1] Lin, J., Tang, J., Tang, H., Yang, S., Xiao, G., & Han, S. (2025). AWQ: Activation-aware Weight Quantization for On-Device LLM Compression and Acceleration. GetMobile: Mobile Comp. and Comm., 28(4), 12–17.
>
>
>
> [2] Steven K. Esser, Jeffrey L. McKinstry, Deepika Bablani, Rathinakumar Appuswamy, & Dharmendra S. Modha (2020). LEARNED STEP SIZE QUANTIZATION. In International Conference on Learning Representations.
>
>
>
> [3] Jungwook Choi, Zhuo Wang, Swagath Venkataramani, Pierce I-Jen Chuang, Vĳayalakshmi Srinivasan, & Kailash Gopalakrishnan. (2018). PACT: Parameterized Clipping Activation for Quantized Neural Networks.
>
>
> [4] Xiao, G., Lin, J., Seznec, M., Wu, H., Demouth, J., & Han, S. (2023). SmoothQuant: accurate and efficient post-training quantization for large language models. In Proceedings of the 40th International Conference on Machine Learning. JMLR.org.
>
>
> [5] D Wu, Q Tang, Y Zhao, M Zhang, Y Fu, and D Zhang. “Easyquant: Post-training quantization via scale optimization.” In: arXiv preprint arXiv:2006.16669 (2020).
>
> [6] Mengwei Xu, Jiawei Liu, Yuanqiang Liu, Felix Xiaozhu Lin, Yunxin Liu, and Xuanzhe Liu. “A First Look at Deep Learning Apps on Smartphones”. In: The World Wide Web Conference. WWW ’19. Association for Computing Machinery, 2019, pp. 2125–2136.

---

> ### Author Response · Authors · 2026-05-02
> **Responses to questions for reviewer 79RK**
>
> **Q1:** Overclaim on hardware efficiency.
>
> The paper argues that existing mixed-precision methods are 'not hardware friendly due to storage and bus bandwidth waste (in related work)', implicitly suggesting that eDQA is a broadly hardware-efficient solution. However, eDQA relies on customized FPGAs, and the total overhead of these custom hardware requirements is never rigorously discussed, nor do the authors consider it necessary to discuss (as said in author's rebuttal). So I think the hardware efficiency claims should be more explicitly scoped, and the method's hardware limitations should be clearly stated. I wonder other reviewers' thought on that.
>
> **R1:**  We will revise the manuscript to moderate the scope of our claims, clarifying that the properties of eDQA suggest the potential for hardware-aware and hardware-friendly approaches, rather than asserting this definitively. We will also emphasize that the current implementation of eDQA presented in the paper is a proof of concept. Its purpose is to demonstrate both the simplicity of integration and that, even with limited design effort, it can outperform NoisyQuant on specialized hardware.
>
> In addition, we will expand the discussion to explicitly acknowledge the limitations of our approach, including hardware-related overheads such as the additional energy cost associated with compression. These points will be incorporated into the revised version to provide a more balanced and transparent evaluation.
>
> ---
>
> **Q2:** Robustness to domain shift.
>
> The authors use AWQ to argue that their offline channel ranking can be treated as a calibration-derived hyperparameter, and thus domain shift is not a fundamental concern. This analogy is misleading. AWQ explicitly claims robustness to domain shift as they say they can avoid overfitting to the calibration set. Crucially, AWQ's scaling is sort of soft operation. In contrast, eDQA's context is about hard bit-level truncation to channels deemed unimportant. Therefore, it is naturally that eDQA is more sensitive to data distribution shift than AWQ. AWQ's explicit robustness claim actually is highlighted in their paper. Can the authors provide further justification?
>
> **R2:** Our work primarily focuses on introducing eDQA, and the analysis of domain shift was not part of the original scope. We note that assuming calibration data to be in-distribution during calibration is conceptually comparable to the common assumption that training data is in-distribution during model training; neither setting is inherently more restrictive than the other. As such, a thorough investigation of domain shift lies beyond the current contribution of this paper.
>
> That said, we agree that this is an important direction. We will acknowledge this limitation more explicitly in the revised manuscript and include a discussion on the potential impact of domain shift on channel ranking. We will also outline plans to investigate this aspect more systematically in future work, including dedicated experiments under domain shift conditions.

---

### Decision · Action_Editor_xBFg · 2026-06-20

**Recommendation:** Accept with minor revision

**Additional Comments:**

There are some remaining concerns raised by Reviewer C3ZN after the rebuttal. These concerns focus on: (1) the lack of a formal error-bound analysis for the proposed shifting-based quantization relative to standard fixed-point arithmetic; (2) insufficient evidence supporting the reported 75% accuracy improvement over NoisyQuant under controlled and standardized experimental settings; and (3) the absence of evaluation on transformer-based vision architectures commonly used in edge deployment, such as Vision Transformer (ViT) and DeiT.

Overall, the reviewers generally agree that the paper makes a valuable contribution to low-bit quantization and edge AI deployment. The outstanding issues are mainly related to clarifying assumptions, appropriately scoping claims, and discussing limitations, rather than any fundamental methodological weaknesses. Therefore, I would like to recommend a minor revision. The authors are encouraged to addressed the above concerns from Reviewer C3ZN, as well as the comment from Reviewer 79RK.

**Audience:**

Yes

**Audience Explanation:**

The paper addresses the important problem of low-bit quantization for efficient edge AI deployment, a topic of broad interest to researchers working on model compression, efficient inference, hardware-aware machine learning, and edge computing systems.

**Claims And Evidence:**

Yes

**Claims Explanation:**

The submission provides convincing empirical evidence across multiple datasets, architectures, and FPGA implementations to support its main claims on accuracy improvement, quantization error reduction, and hardware efficiency.